# Knowledge, attitude, and willingness to perform on-site Cardiopulmonary Resuscitation among individuals trained in public CPR: A cross-sectional survey

ShaoMei Cui, DanJuan Ye, Heng Yang, LiYan Zhang, Lixia Chen*

Department of Nursing, The Fourth Affiliated Hospital of School of Medicine, and international School of Medicine, Internatiinal Institutes of Medicine, Zhejiang University, Yiwu, China

☯ These authors contributed equally to this work.
* chenzheda@zju.edu.cn

## Abstract

### Background

The rescue rate by first responders who have received public Cardiopulmonary resuscitation (CPR) training remains low. While CPR training boosts emergency knowledge and skills among the public, the degree to which this knowledge is retained, along with attitudes and willingness to perform CPR after training, remains elusive. Thus, this study aimed to investigate factors influencing individuals' retention of knowledge, attitude toward CPR, and willingness to perform on-site CPR following training.

### Methods

This cross-sectional study targeted 190 participants from various regions of China who had undergone public CPR training. They completed a questionnaire via online survey between January and February 2024, following CPR training courses.

### Results

Out of 190 distributed questionnaires, 186 were returned and deemed valid, yielding a response rate of 97.9%. The correct response rate for CPR knowledge was merely 39.2%. The majority of respondents had a positive attitude toward on-site CPR, with 86.0% strongly agreeing that "timely CPR can save many lives." 95.7% were willing to perform CPR on a family member. 84.4% of the respondents believe that legal support is the influential factor that affects whether they provide on-site rescue. Factors such as having personal experience in performing CPR on-site, witnessing a cardiac arrest, frequency of CPR training attended in the past 12 months, and educational level significantly influenced (P < 0.05) the mastery of CPR knowledge.

**Data availability statement:** All relevant data are within the manuscript and its Supporting Information files.

**Funding:** The author(s) received no specific funding for this work.

**Competing interests:** The authors have declared that no competing interests exist.

Similarly, these factors, as well as having family members at high risk of cardiogenic sudden death, significantly affected the attitude towards performing CPR on-site ($p < 0.05$).

## Conclusions

Knowledge of CPR remains suboptimal.Although most participants displayed a positive attitude towards performing CPR on-site, their willingness was limited and influenced by various factors. Therefore, organizations offering public CPR training are recommended to implement regular refresher courses, scenario-based simulations, and interactive discussions to mitigate apprehensions and enhance the willingness of trainees for intervention.

## Introduction

Out-of-Hospital Cardiac Arrest (OHCA) is defined as the loss of mechanical cardiac function and systemic circulation outside of hospital settings [1]. Ascribed to its sudden onset, narrow time window for treatment, and poor prognosis, with a global annual incidence of 30–97 per 100,000 and generally low survival rates, it has become a major economic burden on society and a major challenge for global public health [2].

Following the occurrence of OHCA, public bystanders at the scene are generally the last to witness the event and the first to initiate CPR, which can improve the survival rates and neurological outcomes of OHCA patients[3].

However, research data have shown that in developed countries such as Europe and the United States, the implementation rate of bystander cardiopulmonary resuscitation (CPR) ranges from 13% to 82% (58% on average), and the survival rate of patients discharged from the hospital after receiving emergency medical care reaches 8% to 10% [4,5], whereas in China, as the country with the highest incidence of OHCA in the world, more than 544,000 people die of OHCA every year, and the success rate of out-of-hospital resuscitation is only 1%, which is significantly lower than the the level of developed countries [6].This suggests that the dual lack of public first aid ability and willingness to administer first aid has become a core issue limiting the level of life-saving treatment in OHCA.

The critical window for cardiac arrest intervention is the first 4–6 minutes, during which time irreversible damage may occur if not promptly addressed. To enhance the survival rates of OHCA patients, actions taken by initial responders before the arrival of Emergency Medical Services (EMS) are crucial. Thus, their knowledge of emergency procedures and CPR skills can significantly boost their confidence and willingness to intervene. Various countries are dedicated to widespread CPR education initiatives, such as conducting community seminars on CPR and offering public training sessions that include CPR and Heimlich maneuver techniques. Developed countries such as Norway have even incorporated CPR training into mandatory school curricula [7]. Notably, global CPR awareness rates vary from 20% to 70% [8], with the rate of actual CPR implementation at the scene by trained individuals

significantly differing across regions. For instance, the implementation rates are as high as 40.2% and 47.2% in the United States and Europe, respectively [9,10]. On the other hand, the rate of first responders who have received public CPR training and performed CPR is as low as 4.5% in China [11]. Indeed, enhancing the CPR implementation rate among first responders at the scene of a cardiac arrest remains a significant challenge in China.

In China, there is a lack of long-term tracking studies on the effects of public CPR training, and the retention of knowledge, attitudes, and willingness to implement public CPR training are still unclear. Zhejiang Province, as a national demonstration area for public first aid training, has an important demonstration value for the innovation of the training model and evaluation of the implementation effects, but there is a lack of in-depth empirical studies on the trainees in this area.Additionally, this study aimed to analyze factors influencing their behavior in performing CPR to provide a theoretical reference for targeted CPR training. The overarching objective is to increase public awareness, knowledge, and societal accountability regarding emergency interventions for cardiac arrest, thereby increasing on-site CPR rates and enhancing survival outcomes for cardiac arrest patients.

The results of this study will fill the evidence-based gap in the evaluation of the effectiveness of public first aid training in China, provide a theoretical basis for the construction of a "training-practice" transformation model that meets China's national conditions, and have an important practical value for optimizing the design of the CPR training curriculum, improving the first aid legal protection system, and formulating precise intervention strategies.

## Method

### Study design and setting

This study is a cross-sectional study. The design of this study followed the guidelines for reporting observational studies (STROBE).

### Participants

From January 4 to February 22, 2024, participants were recruited using a convenience sampling method from the emergency volunteer training base in Zhejiang Province. **Inclusion Criteria: (1)** Individuals aged 18 years and older. (2) Those who have completed training and received a Zhejiang Province emergency volunteer certificate. (3) Individuals who have provided informed consent to voluntarily participate in this study. To ensure homogeneity among study participants, volunteers trained at other training bases were excluded.

### Sample size

According to sample size estimation methods [12], the sample size is generally 5–10 times the number of study variables. The study included demographic variables of the study subjects (10), knowledge, attitude, and willingness to perform CPR at the scene scale (3 dimensions), a total of 28 variables, and considering 10% invalid questionnaires, the study aimed to recruit at least 156 participants. The current study included 190 study subjects, which meets the above requirements.

### Questionnaire development and validation

The research team designed a survey instrument after reviewing relevant literature on cardiopulmonary resuscitation (CPR) and consulting experts in the field to align with the aim of this study. The content validity and consistency of this questionnaire was validated by five experts, including three senior emergency department physicians specializing in CPR education and resuscitation medicine and two questionnaire experts, and the results showed that this questionnaire has good reliability and validity.

The questionnaire consisted of two sections: a general information form and a scale measuring knowledge, attitudes, and willingness to perform CPR on-site. The former comprised 10 questions covering basic demographic information such

as age, educational level, occupation, and religious beliefs, as well as CPR-specific questions related to family history of cardiogenic sudden death, personal experience with on-site CPR, and witnessing cardiac arrests. The second part assessed knowledge, attitudes, and willingness to perform CPR on-site, containing a total of 28 questions across three dimensions.

The knowledge dimension evaluated the mastery of CPR-related knowledge post-training with 10 questions, wherein correct answers scored 1 point, whereas incorrect answers received none, where higher scores indicate better knowledge retention. The content validity of the knowledge dimension was 0.901 and the Kappa coefficient was 0.515.

The attitude dimension explored the respondents' attitudes towards performing CPR on-site in 15 hypothetical scenarios. To quantify these attitudes, a 5-point Likert scale was used, ranging from "5" (strongly agree) to "1" (strongly disagree), with higher scores indicating more positive attitudes. The content validity of the attitude dimension was 0.924 and the Cronbach's alpha coefficient was 0.913.

The willingness dimension consisted of three questions: "Who would you be willing to perform CPR on?", "Whose opinions would you consider when performing CPR?", and "In what situations would your willingness to perform CPR increase?", thus identifying factors that influence willingness to perform CPR on-site. The content validity of the willingness dimension was 0.905and the Kappa coefficient was 0.621.

## Data collection

The data collection questionnaire was created and distributed using Questionstar software. The survey was conducted after the public had completed CPR training and all participants were informed of the purpose, significance, confidentiality rules and precautions of the study before completing the questionnaire, which was self-completed and submitted online by the participants. To ensure the completeness of the questionnaire, all options were made mandatory questions. To avoid duplication, each IP address could only be entered once. At the end of the survey, the collected questionnaires were evaluated to exclude those completed with obvious regularity and logical errors. A total of 190 questionnaires were collected in this study, and 4 invalid questionnaires with logical errors were excluded to obtain 186 valid questionnaires with a valid recovery rate of 97.9%.

## Ethical considerations

This study was approved by the Human Research Ethics Committee of the hospital (Approval No. K2024039). We certify that the study was performed in accordance with the 1964 declaration of HELSINKI.All the researchers have signed a written informed consent.

## Statistical methods

The data were entered into Excel software and checked by two persons. Data organization and analysis were performed using SPSS software version 24.0. Descriptive statistics were used to describe count data using proportions, while quantitative data were expressed as means ± standard deviations, and differences were evaluated using t-tests and one-way ANOVA. The knowledge of CPR and the attitude toward CPR on-site among those who underwent public training were used as dependent variables, and the variables that had statistical significance ($P < 0.05$) in the results of one-way analysis of variance were used as independent variables, and a multiple linear regression analysis model was established.

## Results

A total of 190 questionnaires were collected, among which 186 were regarded as valid, resulting in a response rate of 97.9%. The proportion of females (59.1%) was higher than that of males (40.9%). The age of respondents ranged from 18 to 66 years, with an average age of 34.68 ± 10.091. Detailed demographic characteristics of the respondents are presented in (Table 1).

**Table 1. Demographic Characteristics of Respondents (n = 186).**

| Characteristics | Sample size N (%) |
|---|---|
| Gender | |
| Female | 110 (59.1) |
| Male | 76 (40.9) |
| Age | |
| 18~38 | 125 (67.2) |
| 39~58 | 57 (30.6) |
| >58 | 4 (2.2) |
| Educational Level | |
| High school and below | 40 (21.5) |
| Associate degree | 57 (30.6) |
| Bachelor's degree and above | 89 (47.9) |
| Occupation | |
| Non-medical related industry | 178 (95.7) |
| Medical related industry | 8 (4.3) |
| Experience in Performing CPR | |
| Present | 50 (26.9) |
| Not present | 136 (73.1) |
| Family Members at High Risk of Cardiogenic Sudden Death | |
| Yes | 25 (13.5) |
| No | 161 (86.5) |
| Witnessed Cardiac Arrest | |
| Yes | 40 (21.5) |
| No | 146 (78.5) |
| Number of CPR Training Sessions Attended in the Past 12 Months | |
| None | 64 (34.4) |
| Once | 99 (53.2) |
| Two or more times | 23 (12.4) |

Respondents achieved a correct response rate of 39.2% in CPR knowledge, with the statement "CPR can be performed without assessing environmental safety, as saving lives is paramount", having the lowest correct response rate, at only 69.9%. The statement "CPR is the simplest and most effective emergency measure for cardiac arrest victims." had the highest correct response rate at 98.4%. Details are listed in (Table 2).

Having direct experience in performing CPR, witnessing a cardiac arrest, the number of CPR training sessions attended in the past 12 months, and the level of education significantly impacted the mastery of CPR knowledge, with a significance level of $P < 0.05$. Similarly, experience in performing CPR, being related to high-risk patients, witnessing a cardiac arrest, the number of CPR training sessions attended in the past 12 months, and the level of education significantly influenced attitudes towards performing CPR on-site, with a significant level of $P < 0.05$, as detailed in (Table 3).

Survey results indicated a predominantly positive attitude among respondents towards performing CPR on-site across the 15 hypothetical scenarios. The distribution of attitudes towards on-site CPR is presented in (Table 4). Interestingly, 86.0% of respondents strongly agreed that "timely CPR can save many lives," and 80.1% strongly agreed that "CPR is valuable because it can save the lives of those experiencing cardiac arrest." Moreover, 82.2% of respondents strongly agreed that "I am eager to learn proper CPR techniques for emergency situations," and 80.1% strongly agreed that

**Table 2. Analysis of Cardiopulmonary Resuscitation Knowledge Mastery.**

| Item | Number Correct | Correct Rate (%) |
|---|---|---|
| The golden rescue time for cardiac arrest victims is only 4–6 minutes; first responders should initiate CPR immediately. | 171 | 91.9 |
| CPR is the simplest and most effective emergency measure for cardiac arrest victims. | 183 | 98.4 |
| Loss of consciousness can also be a sign of cardiac arrest. | 175 | 94.1 |
| The depth of chest compressions should be 5–6 cm. | 177 | 95.2 |
| The chest compression rate should be greater than 100 per minute. | 151 | 81.2 |
| The correct position for adult chest compressions is the midline between the nipples. | 179 | 96.2 |
| The compression-to-breath ratio for adult CPR is 30:2. | 173 | 93.0 |
| The shorter the time to initiate emergency medical measures, the higher the survival rate of the victim. | 181 | 97.3 |
| CPR can be performed without assessing environmental safety, as saving lives is paramount. | 130 | 69.9 |
| CPR should focus only on compressions and does not need mouth-to-mouth resuscitation. | 141 | 75.9 |

"Performing CPR on someone experiencing cardiac arrest does not conflict with my values or religious beliefs." These represent the top four positive attitudes toward performing CPR on-site.

The survey signaled that many respondents were willing to perform CPR on-site for their family members (95.7%). Likewise, there was also a high willingness to perform CPR for friends (85.5%) and colleagues (80.6%). However, the willingness to perform CPR on strangers was significantly lower, at 58.6%. As anticipated, the top three factors that increased the willingness of respondents to perform CPR included "legal protection for first responders" (84.4%), "performing CPR does not lead to personal injury" (78.5%), and "having other bystanders assist in the rescue" (82.8%). Lastly, most respondents (88.7%) prioritized their own thoughts and opinions when performing CPR. Further details are presented in (Table 5).

In the multivariate analysis, higher levels of education, prior experience in performing CPR, witnessing cardiac arrests, and a greater number of CPR training sessions attended in the past 12 months were associated with significantly higher mastery of CPR knowledge ($p < 0.05$). Noteworthily, respondents with higher educational attainment, prior CPR experience, frequent CPR training participation over the last 12 months, family members at high risk of cardiogenic sudden death, and higher scores on CPR knowledge assessment were associated with a more positive attitude towards performing CPR on-site ($p < 0.05$).as detailed in (Table 6).

## Discussion

This study was designed to investigate the retention of knowledge, attitudes toward CPR, and the factors influencing the willingness to perform CPR on-site after public CPR training.

Our survey revealed that the correct mastery level of CPR knowledge among respondents was only 39.3%, which is not significantly different from the 28% in the Arab region [13],which may be linked to the absence of refresher training and the formats used in training sessions[14]. Roughly 34.4% of individuals abstained from participating in CPR training sessions in the past year, and those who attended training one or more times had a better grasp of CPR knowledge. The retention of knowledge and skills was likely to diminish over time,which is consistent with the results of studies in countries such as Nigeria, Pakistan, and Jordan [15,16,17].The Chinese Red Cross and other nonprofit organizations actively encourage public participation in CPR training. However, the lack of retraining for those who have completed initial training

**Table 3. Analysis of differences in knowledge and attitude of CPR on site.**

| variable | knowledge | | | attitude | | |
|---|---|---|---|---|---|---|
| | | Statistical (t/F) | P-value | | Statistical(t/F) | P-value |
| Gender | | 1.586 | 0.115 | | 1.056 | 0.293 |
| Male | 8.21±1.577 | | | 63.61±10.937 | | |
| Female | 8.57±1.499 | | | 61.91±10.659 | | |
| Religious belief | | 0.404 | 0.687 | | 0.605 | 0.553 |
| No | 8.44±1.540 | | | 62.81±10.285 | | |
| Yes | 8.22±1.563 | | | 60.44±15.327 | | |
| Experience in Performing CPR | | 2.823 | 0.005 | | 3.431 | 0.001 |
| Not present | 8.24±1.546 | | | 61.21±11.609 | | |
| Present | 8.94±1.406 | | | 66.09±7.326 | | |
| Family Members at High Risk of Cardiogenic Sudden Death | | 1.348 | 0.179 | | 2.892 | 0.05 |
| No | 8.48±1.505 | | | 61.78±11.207 | | |
| Yes | 8.04±1.719 | | | 66.39±7.533 | | |
| Witnessed Cardiac Arrest | | 2.279 | 0.025 | | 2.583 | 0.011 |
| No | 8.31±1.591 | | | 61.55±11.149 | | |
| Yes | 8.85±1.252 | | | 65.72±8.990 | | |
| Age | | 0.259 | 0.772 | | 0.299 | 0.742 |
| 18-38 | 8.24±1.461 | | | 62.72±10.905 | | |
| 39-58 | 8.30±1.535 | | | 62.09±10.766 | | |
| >58 | 7.75±1.500 | | | 66.25±7.676 | | |
| Number of CPR Training Sessions Attended in the Past 12 Months | | 33.585 | <0.001 | | 10.817 | 0.004 |
| None | 7.56±1.680 | | | 57.84±13.576 | | |
| Once | 9.01±1.111 | | | 64.95±7.718 | | |
| Two or more times | 8.30±1.608 | | | 65.43±9.529 | | |
| Educational Level | | 4.695 | 0.010 | | 9.292 | 0.010 |
| High school and below | 7.73±1.414 | | | 62.95±9.353 | | |
| Associate degree | 8.14±1.620 | | | 58.93±12.477 | | |
| Bachelor's degree and above | 8.55±1.348 | | | 64.80±9.625 | | |

*p<0.05. the $x^2$ test were between each of the groups respectively.

might have contributed to the decline in CPR knowledge retention. Some studies recommend updating CPR knowledge and skills every 11–12 weeks [18].

Furthermore, the survey results unveiled that respondents with higher overall scores in CPR knowledge were more prone to have more positive attitudes towards performing CPR on-site. This may be attributed to the deeper understanding of CPR knowledge and skill accumulation boosting confidence in individuals to act as first responders. In the analysis of the influence of educational level on CPR knowledge mastery, respondents with a bachelor's degree or higher demonstrated superior knowledge retention, in line with previous studies that described a positive correlation between higher educational levels and the mastery of CPR knowledge [19,20,21]. Younger individuals, adept at accessing a broader range of information, including the adverse events related to cardiac arrest, tend to recognize the importance of CPR earlier and are more eager to acquire these lifesaving skills [22]. The survey also exposed that respondent with previous CPR experience or who had witnessed a cardiac arrest exhibited a better grasp of CPR knowledge. As bystanders to CPR,

**Table 4. Distribution of Attitudes Towards Performing CPR On-Site (n, %).**

| Characteristics | 5-Strongly Agree | 4- Mostly Agree | 3- Neutral | 2- Mostly Disagree | 1- Strongly Disagree |
|---|---|---|---|---|---|
| I believe that timely CPR can save many lives. | 160 (86.0) | 22 (11.8) | 4 (2.1) | 0 | 0 |
| Correctly performing CPR in cases of cardiac arrest is easy. | 76 (40.8) | 45 (24.1) | 53 (28.4) | 0 | 12 (6.4) |
| I would perform CPR on a cardiac arrest victim under any circumstances. | 77 (41.3) | 54 (29.0) | 49 (26.3) | 0 | 6 (3.2) |
| CPR is valuable because it can save the lives of those experiencing cardiac arrest. | 149 (80.1) | 34 (18.2) | 3 (1.6) | 0 | 0 |
| I am willing to actively learn proper CPR techniques for emergency situations. | 153 (82.2) | 26 (13.9) | 7 (3.7) | 0 | 0 |
| Performing CPR on someone experiencing cardiac arrest does not conflict with my values or religious beliefs. | 149 (80.1) | 29 (15.5) | 7 (3.7) | 0 | 1 (0.5) |
| If I am the first to aid, I believe I can definitely save the person. | 73 (39.2) | 44 (23.6) | 68 (36.5) | 0 | 1 (0.5) |
| I am not afraid that my attempt to help might cause some harm to the victim. | 81 (43.5) | 48 (25.8) | 53 (28.4) | 0 | 4 (2.1) |
| I am not afraid that my actions in helping might bring about negative consequences for me. | 83 (44.6) | 44 (23.6) | 52 (27.9) | 0 | 7 (3.7) |
| If I perform CPR at the scene, both I and my family will be respected. | 92 (49.4) | 45 (24.1) | 48 (25.8) | 0 | 1 (0.5) |
| I do not think that my actions in helping will make me legally liable. | 81 (43.5) | 47 (25.2) | 54 (29.0) | 0 | 4 (2.1) |
| I am not afraid of contracting a disease (e.g., infectious) by performing CPR at the scene. | 63 (33.8) | 51 (27.4) | 67 (36.0) | 0 | 5 (2.6) |
| If my family knew I performed CPR at the scene, they would be proud of me. | 105 (55.9) | 58 (31.1) | 24 (12.9) | 0 | 0 |
| I would not refuse to perform CPR because there is no barrier device for mouth-to-mouth resuscitation available. | 77 (41.3) | 59 (31.7) | 47 (25.2) | 0 | 3 (1.6) |
| I would feel very guilty if I did not immediately aid someone nearby suffering a cardiac arrest. | 99 (53.2) | 60 (32.2) | 26 (13.9) | 0 | 1 (0.5) |

they understand the critical role of CPR in saving lives and are more motivated to learn during formal training opportunities compared to those who have never been exposed to such emergencies. Furthermore, it is recommended to incorporate scenario-based simulation in CPR public training courses. This teaching method may assist participants in comprehensively grasping rescue knowledge and key points by engaging them in simulated environments [23,24].

Individuals who have undergone public CPR training tend to hold positive attitudes towards performing CPR on-site, a finding corroborated by Mersha AT et al. [25]. It is worthwhile emphasizing that attitudes towards CPR are even more positive abroad, possibly owing to a greater emphasis on CPR education and training. For instance, countries like Pakistan and Slovenia have integrated CPR training into school curricula from an early stage and vigorously promote the importance of CPR [26,27]. Most respondents are aware of the dangers of cardiac arrests after training and recognize the critical role of CPR in saving lives from such events. This shift in values, acquired through training, enhances their willingness to perform CPR on-site [28]. The study also identified a positive correlation between respondents who have family members at high risk of sudden cardiac death and a proactive attitude toward performing CPR on-site. This correlation might stem from the increased awareness of the risk of cardiac arrest while caring for their family members, highlighting the significant impact of cardiac arrests on outcomes for family members [29]. Another study reported that respondents with family members suffering from cardiovascular diseases have less training experience and knowledge of CPR compared to

**Table 5. Factors Influencing Willingness to Perform On-Site CPR (n = 186).**

| Characteristics | Sample size N (%) |
| --- | --- |
| Who would you be willing to perform CPR on? | |
| Family Members | 178 (95.7) |
| Friends | 159 (85.5) |
| Colleagues | 150 (80.6) |
| Strangers | 109 (58.6) |
| Which conditions would increase your willingness to perform CPR on-site? | |
| Legal protection for first responders | 157 (84.4) |
| Performing CPR without risking personal injury | 146 (78.5) |
| Assistance from other bystanders present | 154 (82.8) |
| Only needing to perform chest compressions | 83 (44.6) |
| None of the above conditions would increase my willingness | 7 (3.7) |
| Whose opinions would you consider when performing CPR? | |
| Self | 165 (88.7) |
| Family Members | 107 (57.5) |
| Friends | 86 (46.2) |
| Others | 54 (29.0) |

**Table 6. Multivariate analysis of knowledge and attitude towards CPR on-site.**

| | B | standard error | Standard coefficient | t | p | 95%CI Lower | Upper |
| --- | --- | --- | --- | --- | --- | --- | --- |
| **Knowledge** | | | | | | | |
| Educational Level | 0.476 | 0.127 | 0.255 | 3.734 | 0.000 | 0.224 | 0.727 |
| Experience in Performing CPR | 0.627 | 0.232 | 0.188 | 2.701 | 0.008 | 0.169 | 1.085 |
| Witnessed Cardiac Arrest | 0.573 | 0.245 | 0.160 | 2.341 | 0.020 | 0.090 | 1.055 |
| Number of CPR Training Sessions Attended in the Past 12 Months | 0.464 | 0.157 | 0.204 | 2.958 | 0.004 | 0.154 | 0.773 |
| **attitude** | | | | | | | |
| Educational Level | 2.614 | 0.880 | 0.192 | 2.969 | 0.003 | 0.877 | 4.351 |
| Experience in Performing CPR | 3.694 | 1.740 | 0.158 | 2.123 | 0.035 | 0.260 | 7.127 |
| Witnessed Cardiac Arrest | 1.234 | 1.735 | 0.051 | 0.711 | 0.478 | 2.190 | 4.657 |
| Number of CPR Training Sessions Attended in the Past 12 Months | 2.557 | 1.183 | 0.153 | 2.162 | 0.032 | 0.223 | 4.891 |
| Family Members at High Risk of Cardiogenic Sudden Death | 4.570 | 1.901 | 0.168 | 2.404 | 0.017 | 0.818 | 8.322 |
| Total score of knowledge | 1.700 | 0.472 | 0.253 | 3.603 | 0.000 | 0.749 | 2.632 |

* $p < 0.05$. the $x^2$ test were between each of the groups respectively.

those without such family backgrounds [30]. Consequently, it is recommended to actively conduct targeted CPR training for relatives of high-risk individuals in cardiology wards and outpatient clinics, enhancing their readiness and capability to respond effectively in emergencies.

Most respondents responded that performing CPR correctly is not easy, which may be attributed to the complexity of real-life scenarios where cardiac arrests occur. Indeed, these situations are typically more challenging than those simulated in public training sessions, influenced by the environment, tense atmosphere, and interventions from other

bystanders [31]. Fear of performing CPR also stems from concerns that it might lead to negative outcomes, such as missing key events or facing ostracism or defamation from segments of the public that may not endorse such actions [32]. Additionally, strong disagreement with the statement that CPR would be performed on cardiac arrest victims under any circumstances suggests that even those trained in CPR may hesitate to intervene. Decision-making can be affected by various factors, including the victim's environment, the nature of the injury, and the rescuer's emotional state and attire [33,34]. Therefore, it is crucial that public CPR training programs initially focus on studying cases reported in the media where bystanders have intervened to treat victims. This approach can aid trainees in cultivating a sense of social honor and strengthen their ability to assess the surrounding physical and social environment in case they encounter an individual experiencing cardiac arrest. Thus, training programs should incorporate strategies for making accurate and timely interventions in complex situations while ensuring the personal safety of the rescuer.

Our research findings suggested that most respondents were willing to perform CPR on family members, friends, and colleagues, but their willingness to perform CPR on strangers was notably lower, at 58.6%. This contrasts with other countries, such as Singapore and Korea, where the willingness to perform CPR on strangers after training is 71.6% [35] and 35%, respectively [36]. Additionally, our survey demonstrated that most individuals were more likely to perform CPR in situations where there is legal protection, personal safety is ensured, or there are others available to co-assist, in agreement with the results of Zhou et al. [37]. The Civil Code of the People's Republic of China stipulates that "A rescuer who voluntarily carries out emergency aid that results in harm to the aided person does not bear civil liability." This provision encourages individuals with first aid skills to provide emergency care in critical situations, protected under the law. As skills among the population increase and are highlighted by electronic media, coupled with legal protection, there is a progressive enhancement in their willingness to act. Furthermore, our study indicated that the willingness to perform CPR on-site was primarily influenced by the respondent's personal decision, with minimal impact from family, friends, or others. This underscores the importance of personal autonomy in emergency response decisions. In public CPR training programs, it is crucial to promote altruistic behavior and help participants establish appropriate values and concepts associated with emergency response. Engaging in active discussions about potential concerns and providing direct guidance from instructors on-site can boost confidence and, consequently, increase the willingness of participants to participate in public CPR training [38,39].

## Limitations

Our study has several limitations that should not be overlooked. To begin, the survey was distributed via WeChat to individuals who had undergone public CPR training. The respondents were predominantly young and middle-aged adults, aged 18–38 years, who are likely exposed to the importance of being first responders to cardiac arrests and the necessity of CPR training through online sources. This group's desire to acquire emergency skills for unforeseen situations may lead to more optimistic survey outcomes. Additionally, most respondents completed the questionnaire within one week of receiving CPR training, potentially leading to bias toward more positive responses in the survey results. In the future, surveys should be distributed between six- and twelve-months post-training to more accurately reflect the retention and attitudes of respondents.In the future, we will conduct a multicenter study to expand the sample size to include nationwide recipients of public service CPR training as study subjects, conduct a longitudinal cohort study, and continue to track knowledge and skill retention at 6, 12, and 24 months after training.

## Conclusions

This survey assessed the knowledge, attitudes, and willingness regarding CPR among individuals who participated in public CPR training in Zhejiang, China. Despite recognizing the importance of CPR for cardiac arrest victims and enthusiasm to acquire these lifesaving skills through training, the actual retention of CPR knowledge following training remains low, with personal factors significantly influencing the willingness to perform CPR on-site. Therefore, public CPR training

programs are recommended to include educational strategies that enhance learning, such as scenario-based simulations, open discussions, and lectures on emergency mechanisms and relevant legal information. This approach could reduce rescuers' reluctance and boost trainees' confidence. Additionally, organizing refresher courses to reinforce knowledge and deepen emergency response concepts could further increase trained individuals' willingness to perform CPR on-site.

## Supporting information

**S1 Data. Raw data.**
(XLSX)

**S2 Appendix. Questionnaire.**
(DOCX)

## Acknowledgments

We were grateful to the people who participated in the project questionnaire survey.

## Author contributions

**Data curation:** Danjuan Ye.

**Investigation:** Heng Yang, liyan Zhang.

**Methodology:** lixia Chen.

**Writing – original draft:** shaomei cui.

**Writing – review & editing:** lixia Chen.

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
