## [Decision Letter · Decision Letter 0]

7 Apr 2025

PONE-D-25-06353Knowledge, attitude, and willingness to Perform On-Site Cardiopulmonary Resuscitation Among Individuals Trained in Public CPR: A Cross-Sectional SurveyPLOS ONE

Dear Dr. cui,

Thank you for submitting your manuscript to PLOS ONE. After careful consideration, we feel that it has merit but does not fully meet PLOS ONE’s publication criteria as it currently stands. Therefore, we invite you to submit a revised version of the manuscript that addresses the points raised during the review process.

**Comments from the reviewers:**The abstract should be presented according to the journal's guidelines

The method should state the instrument used (if it is standard and if its reliability and validity in the study is sixteen)

The referencing method should be mentioned according to the journal's format

The necessity of the subject could be better explained

The instrument is not well stated. The reliability and validity method of the instrument that explains this issue does not exist

The method is not well stated

Ethical issues in research should be stated in a focused manner in the final statement of the article. The issue of the ethics committee was stated twice, please state it at the end

The discussion is well-structured and can also be examined with similar countries in terms of health level or gross income allocationIt would be pertinent to repeat this at the 6- and 12-month mark in particular to assess retention of knowledge.

We look forward to receiving your revised manuscript.

Kind regards,

Nik Hisamuddin Nik Ab. Rahman

Academic Editor

PLOS ONE

Journal Requirements:

2. In the online submission form, you indicated that [The data that support the findings of this study are available on request from the corresponding author [LX. C]. The dataare not publicly available due to restrictions, e.g. their containing information that could compromise the privacy ofresearch participants.].

4. We notice that your table are uploaded with the file type 'Figure'. Please amend the file type to 'Table'. Please ensure that each Supporting Information file has a legend listed in the manuscript after the references list.

Reviewers' comments:

Reviewer's Responses to Questions

**Comments to the Author**

1. Is the manuscript technically sound, and do the data support the conclusions?

Reviewer #1: Yes

Reviewer #2: Partly

2. Has the statistical analysis been performed appropriately and rigorously? 

Reviewer #1: I Don't Know

Reviewer #2: No

3. Have the authors made all data underlying the findings in their manuscript fully available?

Reviewer #1: Yes

Reviewer #2: Yes

4. Is the manuscript presented in an intelligible fashion and written in standard English?

Reviewer #1: Yes

Reviewer #2: Yes

5. Review Comments to the Author

Reviewer #1: Thank you for the opportunity to review your paper.

It is an easy-to-read paper with all pertinent points covered and supported by data.

Abstract - sound

Introduction - sound

Methods - straight forward and sound

Results - sound

Limitations - I agree that if this questionnaire was completed within one week of training it would be pertinent to repeat this at the 6- and 12-month mark in particular to assess retention of knowledge.

Conclusions - sound

Reviewer #2: Dear authors,

The abstract should be presented according to the journal's guidelines

The method should state the instrument used (if it is standard and if its reliability and validity in the study is sixteen)

The referencing method should be mentioned according to the journal's format

The necessity of the subject could be better explained

The instrument is not well stated. The reliability and validity method of the instrument that explains this issue does not exist

The method is not well stated

Ethical issues in research should be stated in a focused manner in the final statement of the article. The issue of the ethics committee was stated twice, please state it at the end

The discussion is well-structured and can also be examined with similar countries in terms of health level or gross income allocation

6. PLOS authors have the option to publish the peer review history of their article (what does this mean? ). If published, this will include your full peer review and any attached files.

**Do you want your identity to be public for this peer review?** For information about this choice, including consent withdrawal, please see our Privacy Policy .

Reviewer #1: No

Reviewer #2: No

---

## [Author Response · Author response to Decision Letter 0]

24 Apr 2025

Dear Editor and Reviewers:

We thank all the editors and reviewers for their valuable comments and suggestions.we have carefully revised the manuscript to enhance it’s clarity and facilitate the understanding of the readers.The reviewer comments are laid out below italicized font and specific concerns have been numbered . our response is given in normal font and changes/additions to the manuscript are given in the blue text.We hope that the revision would satisfactorily address the comments and concerns of the editors and reviewers.

Response Letter to Editor’s comments

Comments from the Editor:

Comment 1: When submitting your revision, we need you to address these additional requirements.Please ensure that your manuscript meets PLOS ONE's style requirements, including those for file naming.

Response Thanks for the reminder, the formatting in the revised manuscript has been revised and submitted with reference to the journal's formatting requirements.

Comment 2�In the online submission form, you indicated that [The data that support the findings of this study are available on request from the corresponding author [LX. C]. The dataare not publicly available due to restrictions, e.g. their containing information that could compromise the privacy ofresearch participants.].All PLOS journals now require all data underlying the findings described in their manuscript to be freely available to other researchers.

Response Thanks for the reminder that all data underlying the findings described in this manuscript will be uploaded as supplemental information according to journal requirements.

Comment 3�Your ethics statement should only appear in the Methods section of your manuscript. If your ethics statement is written in any section besides the Methods, please move it to the Methods section and delete it from any other section. Please ensure that your ethics statement is included in your manuscript, as the ethics statement entered into the online submission form will not be published alongside your manuscript.

Response The appearance of the ethical statement twice in the manuscript was an oversight on our part, and we have moved the ethical statement to the “Methods” section and removed the rest.

Comment 4�We notice that your table are uploaded with the file type 'Figure'. Please amend the file type to 'Table'. Please ensure that each Supporting Information file has a legend listed in the manuscript after the references list.

Response We apologize for our oversight. When uploading the manuscript again I will take care and make sure that each supporting information file has a legend listed after the reference list of the manuscript.

Response Letter to Reviewer’s comments

Comments from the Reviewers

Comment 1�The abstract should be presented according to the journal's guidelines

Response� Thank you very much for your valuable advice. I apologize for our oversight. We carefully reviewed the PLOS One submission guidelines and rewrote the abstract section of the manuscript to describe the main objectives of the study, the methodology of the study, and to summarize the results and significance of the study, and revised the abstract section to be no more than 300 words in length.

The revised abstract is as follows:

Background

The rescue rate by first responders who have received public Cardiopulmonary resuscitation (CPR) training remains low. While CPR training boosts emergency knowledge and skills among the public, the degree to which this knowledge is retained, along with attitudes and willingness to perform CPR after training, remains elusive. Thus, this study aimed to investigate factors influencing individuals' retention of knowledge, attitude toward CPR, and willingness to perform on-site CPR following training.

Methods

This cross-sectional study targeted 190 participants from various regions of China who had undergone public CPR training. They completed a questionnaire via online survey between January and February 2024, following CPR training courses.

Results

Out of 190 distributed questionnaires, 186 were returned and deemed valid, yielding a response rate of 97.9%. The correct response rate for CPR knowledge was merely 39.2%. The majority of respondents had a positive attitude toward on-site CPR, with 86.0% strongly agreeing that "timely CPR can save many lives." 95.7% were willing to perform CPR on a family member. 84.4% of the respondents believe that legal support is the influential factor that affects whether they provide on-site rescue. Factors such as having personal experience in performing CPR on-site, witnessing a cardiac arrest, frequency of CPR training attended in the past 12 months, and educational level significantly influenced (P<0.05) the mastery of CPR knowledge. Similarly, these factors, as well as having family members at high risk of cardiogenic sudden death, significantly affected the attitude towards performing CPR on-site (p<0.05).

Conclusions

Knowledge of CPR remains suboptimal.Although most participants displayed a positive attitude towards performing CPR on-site, their willingness was limited and influenced by various factors. Therefore, organizations offering public CPR training are recommended to implement regular refresher courses, scenario-based simulations, and interactive discussions to mitigate apprehensions and enhance the willingness of trainees for intervention.

Word Count: 298

(Page 2-3)

Comment 2�The method should state the instrument used (if it is standard and if its reliability and validity in the study is sixteen)

Response�we would like to express our sincere gratitude to the reviewer for your insightful comments and valuable feedback.We described the research instrument used in this study in the Methods section, we used a questionnaire and described the two parts of the questionnaire in detail. The first part was a general information questionnaire and the second part was a survey on knowledge, attitude and willingness to perform CPR on site. The second part, among others, was divided into three dimensions: knowledge of CPR, attitude toward performing CPR on-site, and willingness to perform CPR on-site, and tests of reliability and validity were added to these three dimensions.

The modified part is as follows:

Questionnaire development and validation

The research team designed a survey instrument after reviewing relevant literature on cardiopulmonary resuscitation (CPR) and consulting experts in the field to align with the aim of this study. The content validity and consistency of this questionnaire was validated by five experts, including three senior emergency department physicians specializing in CPR education and resuscitation medicine and two questionnaire experts, and the results showed that this questionnaire has good reliability and validity.

The questionnaire consisted of two sections: a general information form and a scale measuring knowledge, attitudes, and willingness to perform CPR on-site. The former comprised 10 questions covering basic demographic information such as age, educational level, occupation, and religious beliefs, as well as CPR-specific questions related to family history of cardiogenic sudden death, personal experience with on-site CPR, and witnessing cardiac arrests. The second part assessed knowledge, attitudes, and willingness to perform CPR on-site, containing a total of 28 questions across three dimensions.

The knowledge dimension evaluated the mastery of CPR-related knowledge post-training with 10 questions, wherein correct answers scored 1 point, whereas incorrect answers received none, where higher scores indicate better knowledge retention. The content validity of the knowledge dimension was 0.901 and the Kappa coefficient was 0.515.

The attitude dimension explored the respondents' attitudes towards performing CPR on-site in 15 hypothetical scenarios. To quantify these attitudes, a 5-point Likert scale was used, ranging from "5" (strongly agree) to "1" (strongly disagree), with higher scores indicating more positive attitudes. The content validity of the attitude dimension was 0.924 and the Cronbach's alpha coefficient was 0.913.

The willingness dimension consisted of three questions: "Who would you be willing to perform CPR on?", "Whose opinions would you consider when performing CPR?", and "In what situations would your willingness to perform CPR increase?", thus identifying factors that influence willingness to perform CPR on-site. The content validity of the willingness dimension was 0.905and the Kappa coefficient was 0.621.

(Page7,line20 to Page9,Line 8)

Comment 3�The referencing method should be mentioned according to the journal's format

Response Thank you very much for your valuable advice and I apologize for our oversight. We have carefully reviewed the submission guidelines for PLOS One, which uses the reference style prescribed by the International Committee of Medical Journal Editors (ICMJE), also known as the “Vancouver” style. We have completed the changes to the reference format.

The revised reference format is as follows:

1.Myat A, Song KJ, Rea T. Out-of-hospital cardiac arrest: current concepts. Lancet.2018;391(10124):970-9.doi:10.1016/S0140-6736(18)30472-0. PMID: 29536860.

Page20�line4 to Page25�line19

Comment 4�The necessity of the subject could be better explained

Response Thank you for your suggestion. We rephrased the statements about the importance and necessity of the study in the introduction section.

The revised introduction is as follows:

Introduction

Out-of-Hospital Cardiac Arrest (OHCA) is defined as the loss of mechanical cardiac function and systemic circulation outside of hospital settings [1]. Ascribed to its sudden onset, narrow time window for treatment, and poor prognosis, with a global annual incidence of 30-97 per 100,000 and generally low survival rates, it has become a major economic burden on society and a major challenge for global public health [2].

Following the occurrence of OHCA, public bystanders at the scene are generally the last to witness the event and the first to initiate CPR, which can improve the survival rates and neurological outcomes of OHCA patients[3].

However, research data have shown that in developed countries such as Europe and the United States, the implementation rate of bystander cardiopulmonary resuscitation (CPR) ranges from 13% to 82% (58% on average), and the survival rate of patients discharged from the hospital after receiving emergency medical care reaches 8% to 10% [4, 5], whereas in China, as the country with the highest incidence of OHCA in the world, more than 544,000 people die of OHCA every year, and the success rate of out-of-hospital resuscitation is only 1%, which is significantly lower than the the level of developed countries [6].This suggests that the dual lack of public first aid ability and willingness to administer first aid has become a core issue limiting the level of life-saving treatment in OHCA.

The critical window for cardiac arrest intervention is the first 4-6 minutes, during which time irreversible damage may occur if not promptly addressed. To enhance the survival rates of OHCA patients, actions taken by initial responders before the arrival of Emergency Medical Services (EMS) are crucial. Thus, their knowledge of emergency procedures and CPR skills can significantly boost their confidence and willingness to intervene. Various countries are dedicated to widespread CPR education initiatives, such as conducting community seminars on CPR and offering public training sessions that include CPR and Heimlich maneuver techniques. Developed countries such as Norway have even incorporated CPR training into mandatory school curricula [7]. Notably, global CPR awareness rates vary from 20% to 70% [8], with the rate of actual CPR implementation at the scene by trained individuals significantly differing across regions. For instance, the implementation rates are as high as 40.2% and 47.2% in the United States and Europe, respectively [9,10]. On the other hand, the rate of first responders who have received public CPR training and performed CPR is as low as 4.5% in China [11]. Indeed, enhancing the CPR implementation rate among first responders at the scene of a cardiac arrest remains a significant challenge in China.

In China, there is a lack of long-term tracking studies on the effects of public CPR training, and the retention of knowledge, attitudes, and willingness to implement public CPR training are still unclear. Zhejiang Province, as a national demonstration area for public first aid training, has an important demonstration value for the innovation of the training model and evaluation of the implementation effects, but there is a lack of in-depth empirical studies on the trainees in this area.Additionally, this study aimed to analyze factors influencing their behavior in performing CPR to provide a theoretical reference for targeted CPR training. The overarching objective is to increase public awareness, knowledge, and societal accountability regarding emergency interventions for cardiac arrest, thereby increasing on-site CPR rates and enhancing survival outcomes for cardiac arrest patients.

The results of this study will fill the evidence-based gap in the evaluation of the effectiveness of public first aid training in China, provide a theoretical basis for the construction of a “training-practice” transformation model that meets China's national conditions, and have an important practical value for optimizing the design of the CPR training curriculum, improving the first aid legal protection system, and formulating precise intervention strategies.

(Page4,line5 to Page6,line20)

Comment 5�The instrument is not well stated. The reliability and validity method of the instrument that explains this issue does not exist

Response�we would like to express our sincere gratitude to the reviewer for your insightful comments and valuable feedback.We described the research instrument used in this study in the Methods section, we used a questionnaire and described the two parts of the questionnaire in detail. The first part was a general information questionnaire and the second part was a survey on knowledge, attitude and willingness to perform CPR on site. The second part, among others, was divided into three dimensions: knowledge of CPR, attitude toward performing CPR on-site, and willingness to perform CPR on-site, and tests of reliability and validity were added to these three dimensions.

The modified part is as follows:

Questionnaire development and validation

The research team designed a survey instrument after reviewing relevant literature on cardiopulmonary resuscitation (CPR) and consulting experts in the field to align with the aim of this study. The content validity and consistency of this questionnaire was validated by five experts, including three senior emergency department physicians specializing in CPR education and resuscitation medicine and two questionnaire experts, and the results showed that this questionnaire has good reliability and validity.

The questionnaire consisted of two sections: a general information form and a scale measuring knowledge, attitudes, and willingness to perform CPR on-site. The former comprised 10 questions covering basic demographic information such as age, educational level, occupation, and religious beliefs, as well as CPR-specific questions related to family history of cardiogenic sudden death, personal experience with on-site CPR, and witnessing cardiac arrests. The second part assessed knowledge, attitudes, and willingness to perform CPR on-site, containing a total of 28 questions across three dimensions.

The knowledge dimension evaluated the mastery of CPR-related knowledge post-training with 10 questions, wherein correct answers scored 1 point, whereas incorrect answers received none, where higher scores indicate better knowledge retention. The content validity of the knowledge dimension was 0.901 and the Kappa coefficient was 0.515.

The attitude dimension explored the respondents' attitudes towards performing CPR on-site in 15 hypothetical scenarios. To quantify these attitudes, a 5-point Likert scale was used, ranging from "5" (strongly agree) to "1" (strongly dis

---

## [Editor Report · Decision Letter 1]

12 May 2025

Knowledge, attitude, and willingness to Perform On-Site Cardiopulmonary Resuscitation Among Individuals Trained in Public CPR: A Cross-Sectional Survey

PONE-D-25-06353R1

Dear Dr. shaomei cui%,

We’re pleased to inform you that your manuscript has been judged scientifically suitable for publication and will be formally accepted for publication once it meets all outstanding technical requirements.

Kind regards,

Nik Hisamuddin Nik Ab. Rahman

Academic Editor

PLOS ONE
---

## [Editor Report · Acceptance letter]

PONE-D-25-06353R1

PLOS ONE

Dear Dr. cui,

I'm pleased to inform you that your manuscript has been deemed suitable for publication in PLOS ONE. Congratulations! Your manuscript is now being handed over to our production team.

Kind regards,

on behalf of

Professor Dr Nik Hisamuddin Nik Ab. Rahman

Academic Editor

PLOS ONE